# Use of Differential Entropy for Automated Emotion Recognition in a Virtual Reality Environment with EEG Signals

**DOI:** 10.3390/diagnostics12102508

**Published:** 2022-10-16

**Authors:** Hakan Uyanık, Salih Taha A. Ozcelik, Zeynep Bala Duranay, Abdulkadir Sengur, U. Rajendra Acharya

**Affiliations:** 1Electrical-Electronics Engineering Department, Engineering Faculty, Munzur University, Tunceli 62000, Turkey; 2Electrical-Electronics Engineering Department, Engineering Faculty, Bingol University, Bingol 12000, Turkey; 3Electrical-Electronics Engineering Department, Technology Faculty, Firat University, Elazig 23119, Turkey; 4Department of Electronics and Computer Engineering, Ngee Ann Polytechnic, Singapore 599489, Singapore; 5Biomedical Engineering, School of Science and Technology, SUSS University, Singapore 599494, Singapore; 6Biomedical Informatics and Medical Engineering, Asia University, Taichung 413305, Taiwan

**Keywords:** EEG signal, virtual reality (VR)-based emotions, differential entropy, SVM

## Abstract

Emotion recognition is one of the most important issues in human–computer interaction (HCI), neuroscience, and psychology fields. It is generally accepted that emotion recognition with neural data such as electroencephalography (EEG) signals, functional magnetic resonance imaging (fMRI), and near-infrared spectroscopy (NIRS) is better than other emotion detection methods such as speech, mimics, body language, facial expressions, etc., in terms of reliability and accuracy. In particular, EEG signals are bioelectrical signals that are frequently used because of the many advantages they offer in the field of emotion recognition. This study proposes an improved approach for EEG-based emotion recognition on a publicly available newly published dataset, VREED. Differential entropy (DE) features were extracted from four wavebands (theta 4–8 Hz, alpha 8–13 Hz, beta 13–30 Hz, and gamma 30–49 Hz) to classify two emotional states (positive/negative). Five classifiers, namely Support Vector Machine (SVM), k-Nearest Neighbor (kNN), Naïve Bayesian (NB), Decision Tree (DT), and Logistic Regression (LR) were employed with DE features for the automated classification of two emotional states. In this work, we obtained the best average accuracy of 76.22% ± 2.06 with the SVM classifier in the classification of two states. Moreover, we observed from the results that the highest average accuracy score was produced with the gamma band, as previously reported in studies in EEG-based emotion recognition.

## 1. Introduction

Two kinds of phenomena, namely thought/cognition and emotion, comprise our inner spiritual life. These two concepts are interrelated and are personal experiences that cannot be directly observed by others [1], and these two phenomena also interact with environmental events [2,3]. Therefore, disciplines such as neuroscience, cognitive science, and psychology investigate the effects and consequences of these phenomena on human life.

Two emotion models are widely used in emotion recognition [4]. These are the discrete emotion models [5] and the bi-directional emotion models [6]. While there are emotions such as fear, sadness, happiness, anger, and disgust in the discrete emotion model, in the bi-dimensional emotion model, emotions are found in a multidimensional model on the valence and arousal scales. A more general and simplified version of multidimensional model, a four-category structure, is frequently used in the literature. Within this structure, the emotion categories are low arousal-low valence (LALV), low arousal-high valence (LAHV), high arousal-low valence (HALV), and high arousal-high valence. In this study, we used a discrete emotion model.

Human emotions can be detected using speech signals [7], facial expressions [8], body language [9], electroencephalography (EEG) [10], posture, gesture, etc. [11]. Among these emotion detection approaches, the detection with EEG signals is the approach that obtained the safest results. Because EEG signals are autonomic nervous system (ANS) signals obtained directly from the brain and due to which emotion occurs [12]. In other methods, people may reflect their instant feelings differently. For example, people can adjust their facial expressions to positive in a negative emotional status. This situation cannot be changed by physiological signals (EEG, electromyography (EMG), electrocardiography (ECG), temperature, etc.).

To date, many researchers have performed emotion recognition on various EEG datasets [13,14,15,16]. These datasets are mostly created by recording the EEG signals that occur on the subjects by applying stimuli such as sound or images (picture/video) to the subjects in a certain procedure. The images used in these created datasets mostly have 2-dimensional stimuli, such as in the most popular DEAP [17], SEED [18], and MAHNOB-HCI [19] datasets. For example, Ari et al. [10] used the GAMEEMO [20] dataset in their work, where a novel data augmentation approach (the Extreme Learning Machine Wavelet Auto Encoder) was proposed for efficient human emotion recognition. Authors initially converted the EEG signals to the EEG images using continuous wavelet transform (CWT). Moreover, the obtained images were augmented by using the proposed data augmentation method. Lastly, the fine-tuning of the ResNet-18 was used for classification and reported a classification accuracy of 99.6%. Demir et al. [21] used pre-trained deep convolutional neural network (DCNN) models for feature extraction from the EEG signals. Five pre-trained DCNN models were considered, and a manual channel and rhythm selection method was used to find the most convenient channel and rhythm for accurate emotion recognition. The authors used the DEAP dataset in their work and obtained a 98.93% accuracy score. Tuncer et al. [22] used tetromino features and an SVM classifier for EEG-based emotion classification. The authors decomposed the input EEG signal into sub-bands using the discrete wavelet transform (DWT), and the tetromino features were extracted for each sub-band. They reported 100%, 100%, and 99% classification accuracies for Dreamer, Gameemo, and DEAP datasets, respectively. Ismael et al. [23] used a two-stepped approach for EEG-based emotion classification. Low pass filtering was employed for noise elimination, and bandpass filtering was employed for rhythm extraction. The best-performing EEG channels for each rhythm were determined using the KNN classifier, wavelet-based entropy characteristics, and fractal dimension-based features. Then, the best five EEG channels were considered for majority voting to obtain the final prediction of each EEG rhythm. The DEAP dataset was considered in the proposed work and obtained the classification accuracy of 86.3% for HV vs. LV and 85.0% for HA vs. LA. Joshi et al. [24] proposed an EEG-based emotion recognition framework where Linear Formulation of Differential Entropy ($LF-D_{f}E$) was developed for feature extraction, and the Bidirectional Long Short-Term Memory (BiLSTM) approach was used for the classification of the $LF-D_{f}E$ features. Moreover, the $LF-D_{f}E$ features were used to characterize the nonlinearity and non-Gaussianity of the input EEG signal. They obtained 80.64% and 76.75% accuracy scores using SEED and DEAP datasets, respectively. Krishna et al. [25] used Tunable-Q Wavelet Transform (TQWT) and graph-regularized extreme learning machine to classify different emotion EEG signals. The authors decomposed the input EEG signals into sub-bands with TQWT. They extracted various statistical features such as root mean square, log detector, clearance, crest, shape, activity, and mobility. The authors obtained a classification accuracy of 87.1%. Yin et al. [26] proposed a model based on deep learning for EEG-based emotion classification. The authors used a window to extract a feature cube calculated using the DE from the EEG signals. Then, a deep learning model was employed to classify the obtained features into emotion labels. The experimental results for the subject-dependent experimental works revealed 90.45% and 90.60% accuracy scores, and subject-independent experiments obtained 84.81% and 85.27% classification accuracy scores for valence and arousal classification, respectively. Chen et al. [27] used an EEG-based emotion recognition system using the Library for Support Vector Machines (LIBSVMs) [28] classifier that is an integrated software for support vector classification (C-SVC, nu-SVC), regression (epsilon-SVR, nu-SVR), and distribution prediction (single class SVM). It also supports multi-class classification. The authors initially used the Lempel–Ziv complexity, wavelet coefficients, empirical mode decomposition, and intrinsic mode functions for feature extraction, and LIBSVM was used to classify the extracted features. The authors achieved a 74.88% accuracy score for arousal, and 82.63% accuracy score was obtained for valence. Gao et al. [29] proposed a new multi-featured fusion network consisting of spatial and temporal neural network structures for emotion recognition with EEG signals. Both time domain and frequency domain features were extracted. Moreover, the authors developed a GoogleNet-inspired convolutional neural network (CNN) model to capture the intrinsic spatial relationship between EEG electrodes and contextual information. The DEAP dataset was used in the experimental works, and 80.52% and 75.22% classification accuracies were reported for valence and arousal classes, respectively. Xing et al. [30] used an emotion recognition model consisting of a linear EEG mixing and an emotion timing models. They used the Stack AutoEncoder to generate and solve the linear EEG mixing model and the Long Short-Term Memory Recurrent Neural Network (LSTM RNN) for the emotion timing model. The authors used the DEAP dataset and achieved an 81.10% accuracy score for the valence class and 74.38% accuracy score for the arousal class.

Few researchers have recently constructed new datasets with high-tech 3D virtual reality images. For example, Morales et al. [31] implemented an emotion recognition system that can be applied in three dimensions or real environments to evoke emotional states. They constructed a dataset where both EEG and electrocardiogram (ECG) signals were recorded from 60 participants using four virtual rooms. Two of the four rooms emphasized positive emotions, and the other two rooms emphasized negative emotions. Power spectral density (PSD), phase coupling, and heart rate variability were extracted features from the input signals, and an SVM classifier was used for classification. Experimental works revealed that the proposed scheme produced a 71.21% accuracy score for the valence class and a 75% accuracy score for the arousal class. Suhami et al. [32] used virtual reality glasses as stimuli to classify human emotions (happy, scared, calm, and bored) with an EEG device. As a result of the experiments they carried out with SVM, they reached a classification accuracy of 85.01% for the four-class emotion recognition, including the more difficult classification process, such as inter-subject classification.

Linear interpolation and spline interpolation were used for feature extraction, and the SVM classifier was used for classification. Experimental works produced an 85.01% accuracy score for the classification of the four emotions. The Differential Entropy (DE) method is frequently preferred in EEG emotion recognition [33,34,35,36]. In their study [37], Zheng et al. applied six different feature extraction methods (PSD, DE, Differential Asymmetry (DASM), Rational Asymmetry, Asymmetry (RASM), Differential Caudality) for emotion recognition. The authors used DEAP and SEED datasets with four different classification methods: KNN, Logistic Regression (LR), SVM, and graph-regularized extreme learning machine (GELM). In their work, DE features obtained higher accuracy and lower standard deviation values than the other five features. The best average accuracies of 69.67% and 91.07% were obtained using discriminative GELM with DE features for DEAP and SEED datasets, respectively.

This paper proposes an EEG-based emotion classification approach for a 3D VR-based dataset [38]. In the proposed approach, the input EEG signals were initially preprocessed with MATLAB toolbox EEGLAB [39] for noise and baseline removals. Moreover, the independent component analysis was used to prune the EEG signals. Then, a rhythm extraction procedure based on wavelet decomposition was employed to obtain the four rhythms, theta, alpha, beta, and gamma. The theta band oscillates between 4 and 8 Hz from these four bands. The activity of this band was heightened in moments of meditation and drowsiness. The alpha band, called the basic rhythm, consists of frequencies between 8 and 13 Hz. The activity of this band is high during awake and eyes open. It weakens especially when visual attention and mental effort are required during motor activities. Its activity is high in the middle part of the head. The beta band oscillates between 13 and 30 Hz. It becomes more evident during active thinking and concentration. The gamma band covers oscillations of 30 Hz and above. It is related to information processing and the onset of voluntary movements [40].

Experiments were carried out with the properties obtained from these bands. The Differential Entropy (DE) features are extracted from all 59 channels of each rhythm. SVM, kNN, NB, DT, and LR classifiers are used in the classification stage of the proposed scheme. The VREED dataset, which contains 19 subjects, is considered in the experimental works, and various evaluation metrics are used to evaluate the performance of the proposed method. Hold-out validation method, 70% of each subject is used for training, and the other 30% of each subject is used for testing purposes. We obtained the maximum average accuracy of 76.22% ± 2.06 with the SVM classifier. The original contributions of this work are as follows:To the best of our knowledge, we obtained the highest classification accuracy using the VREED dataset;This is the first-time Differential Entropy has been used for the VREED dataset.

The rest of the paper is organized as follows: In Section 2, the proposed scheme and its application steps are provided. Moreover, the theory of the considered methods and the description of the dataset is provided in Section 2. In Section 3, the experimental results are interpreted in detail. In Section 4, the advantages and disadvantages of the proposed method are emphasized. Finally, the paper is concluded in Section 5.

## 2. Proposed Method

The methodology of the proposed scheme is illustrated in Figure 1. As seen in Figure 1, the input EEG signals are initially decomposed into four sub-bands for rhythm extraction. These sub-bands are theta, alpha, beta, and gamma. For the rhythm extraction, the WT used the ‘db8’ mother wavelet. The 5-level decomposition is employed, and the delta band is not considered due to eliminating noises such as pulses, neck movement, and eye blinking [41]. After the rhythms are extracted, DE is used to extract a bunch of features from all channels of each EEG rhythm. As mentioned in [42], the logarithm of the energy spectrum of a fixed-length EEG signal is equal to the DE of the signal [29].

Hence, the DE of the alpha, beta, gamma, and theta bands of the EEG signal are used to classify the negative and positive emotions. Each rhythm contains 59 channels, so the DE feature set is a 59-dimensional feature vector. The SVM, kNN, NB, DT, and LR classifiers are used in the classification stage of the proposed scheme. The classification accuracy is used as the performance evaluation metric.

### 2.1. Wavelet Transform

A signal is decomposed into a series of mutually orthogonal wavelet basis functions using the wavelet transform (*WT*) [4]. These functions vary from sinusoidal basis functions in which they are spatially localized, meaning that they are nonzero only across a portion of the whole signal duration. Wavelet functions are also dilated, translated, and scaled variants of the mother wavelet, a common function *ψ*. The *WT* is invertible, much as Fourier analysis; hence the original signal can be fully retrieved from its *WT* form. The continuous *WT* (*CWT*) is defined as:(1)$$CWT\left(a,b\right)=\frac{1}{\sqrt{\left|a\right|}}\int f\left(t\right)\psi \left(\frac{t-b}{a}\right)\partial t$$
where *a* and *b* are the scale and translation parameters, respectively. *ψ* and *f*(*t*) are the wavelet function and given signal, respectively. The discrete *WT* (*DWT*) is defined as:(2)$$DWT\left(m,n\right)=\int f\left(t\right)\psi_{m,n}\left(t\right)\partial t$$
where *m* and *n* are the scaling and translation constants, respectively. In this work, we used *DWT* for our analysis.

### 2.2. Differential Entropy (DE)

The concept of DE is the equivalent of the concept of entropy for continuous distributions in Shannon’s original paper [43]. The DE is used to measure the complexity of continuous and random variables. However, it measures the relative uncertainty, or changes in the uncertainty, rather than calculating an absolute measure of uncertainty [44].

Consider continuous time random variable *X* and $p_{X}\left(x\right)$ is the probability density function (PDF) of *X*, the DE of *X* is calculated as follows:(3)$$h_{X}=-\overset{.}{\underset{S}{\int }}p_{X}\left(x\right)\log\left(p_{X}\left(x\right)\right)dx$$
where *S* = {x|$p_{X}\left(x\right)$ > 0} is the support set of X [44]. As the random variable fits the Gaussian distribution $N\left(\mu ,\;\sigma^{2}\right)$, the DE of that variable is calculated as follows:(4)$$p\left(x\right)=\frac{1}{\sqrt{2\pi \sigma }}\;e^{-\frac{x^{2}+\mu^{2}}{2\sigma^{2}}}\;cosh\left(\frac{\mu x}{\sigma^{2}}\right)$$

Then the DE can be calculated as:(5)$$h_{X}=\overset{+\infty }{\underset{-\infty }{\int }}p\left(x\right)\ln\left(p\left(x\right)\right)dx$$
(6)$$h_{X}=\frac{1}{2}\;\log_{2}\left(2\pi e\sigma^{2}\right)+L\left(\frac{\mu }{\sigma }\right)$$
where L(⋅) is the function of μ/σ which goes from 0 to 1 (ln2), *e* is Euler’s constant, and σ is the standard deviation of *x*. Figure 2 shows the DE values for positive and negative emotions for the EEG signal of subject-12. The columns of Figure 2 show the DE values for each rhythm for positive and negative emotions.

### 2.3. Dataset

The dataset used in the study was created by watching 4 s long 60 VR videos of three types recorded in 3D by the Shanghai Film Academy with the help of VR glasses. These video types are positive, negative, and neutral. The dataset was initially created with a group of 15 men and 10 women with a mean age of 22.92 and a standard deviation of 1.38 for these ages. The following steps were applied in the emotion elicitation protocol stage:

As provided in Figure 3, 60 videos from three types were randomly selected and divided into two groups, 20 positive–10 negative and 20 negative–10 neutral. Researchers who created this dataset did not include negative and positive images in the same video group since VR images have a much higher stimulating effect than normal two-dimensional images. Thus, interference between different emotions was prevented. Each subject was shown two different groups of videos, and the videos in these groups were in random order. Participants were provided a 3 min relaxation time before starting the experiment. After a warning sign of 3 s, the video was played. In this three-second warning sign, the electrical activity in the brain is marked as a baseline in the dataset. At the end of the videos, a relaxation time was provided to the subjects. At the end of each video watched, the subjects were asked to evaluate the induction of the videos on the subjects’ emotions. The videos in two groups were shown to each subject twice with the above-mentioned steps. In other words, 120 EEG signal data were collected from each subject.

These recordings were recorded with a 64-channel wireless EEG recorder with a sampling frequency of 1000 Hz, with the impedance of the probes below 5 kΩ. A total of 59 of these 64 channels, corresponding to five different regions of the brain: occipital, parietal, frontal, right temporal, and left temporal, were placed according to the expanded international 10-20 system (10-10 systems) [45] as seen in Figure 4.

After all the recording processes were completed, these data were preprocessed with the help of the EEGLAB Toolbox [39]. Low-quality recordings were deleted during this preprocessing. After this deletion, a dataset of 19 people with a mean age of 22.84 and a standard deviation of 1.50 for these ages was formed. Then, the artifacts and power frequency interference of these data were removed, and the baseline parts were marked.

## 3. Experimental Results

The experiments were conducted on MATLAB. Initially, EEGLAB was used to extract the epochs for each subject. Then, each EEG signal was used to discriminate two-class emotion recognition. Finally, hold-out cross-validation was utilized by dividing the samples of each participant into a 70% training set and a 30% test set while maintaining a roughly constant percentage of each class in each set compared with the original data. The classification process was performed ten times, and the average classification accuracy was computed to lessen the unpredictability introduced by the random partition of the dataset. As mentioned earlier, various machine learning architectures were used in the classification stage of the proposed method. The parameters of these classifiers were tuned by using a hyperparameter optimization algorithm.

Table 1 shows the average accuracy scores obtained with the DE features and the mentioned machine learning classifiers for each rhythm and concatenated all rhythms, respectively. For example, Table 1 shows that using the SVM classifier, 61.0773%, 66.6826%, 70.9104%, and 73.9968% average accuracies were obtained using the DE features for theta, alpha, beta, and gamma rhythms, respectively. Moreover, we obtained 58.0034%, 65.4013%, 68.8021%, and 72.9261% average accuracies for the theta, alpha, beta, and gamma rhythms, respectively. Similarly, we obtained average accuracy scores for DT, NB, and LR in the third, fourth, and fifth rows of Table 1, respectively. The last column of Table 1 also shows the obtained average accuracy scores for each classifier by concatenating all the features from all rhythms. It can be noted that the SVM produced its best average accuracy score of 76.2236% by concatenating all rhythms, where the recorded average accuracy score was 76.2236%.

Similarly, the kNN classifier produced the best average accuracy score of 72.9261% with the gamma rhythm. Classification accuracy of 72.6460% was obtained for all rhythms. The classifiers NB, DT, and LR, obtained average accuracy scores of 59.7951%, 61.5310%, and 64.0111%, respectively. It can be noted from the obtained results that the gamma band produced the best average accuracy scores with all classifiers.

Figure 5 shows the cumulative confusion matrices obtained using the DE features and SVM classifier. The blue cells show the true average classifications for both positive and negative classes, and the orange regions show the false classifications for each class. For example, all rhythm concatenations obtained 78.9% and 73.6% average accuracy scores for positive and negative classes, respectively.

In Figure 6, the obtained cumulative confusion matrices for each rhythm and concatenated all rhythms for the kNN classifier are provided. As the gamma rhythm produced the best achievement for the kNN classifier, its cumulative confusion matrices were observed, and the true classification accuracies for positive and negative classes were 74.5% and 73.5%, respectively. Conversely, the false positive and negative accuracy scores were 25.5% and 26.5%, respectively.

Similarly, Figure 7 shows the obtained confusion matrices for the NB classifier. Again, the true classification accuracies for positive and negative classes for the gamma rhythm were 60.4% and 59.2%, respectively.

Moreover, the false classifications for positive and negative classes were 39.6% and 40.8%, respectively. In Figure 8, the obtained cumulative confusion matrices for each rhythm and all rhythms using the DT classifier were provided. It can be noted that the gamma rhythm produced the best achievement using the DT classifier. While the true classification accuracies for positive and negative classes were 61.1% and 62%, respectively. The false positive and negative accuracy scores were 38.9% and 38%, respectively.

Figure 9 shows the obtained cumulative confusion matrices for each rhythm and all rhythms using LR classifier. Similar to the previous results, the gamma rhythm produced the best achievement with this classifier. The true positive and true negative rates obtained were 63.5% and 64.6%, respectively. The false positive and negative accuracy scores obtained were 36.5% and 35.4%, respectively.

## 4. Discussion

This paper proposed an improved approach for 3D-VR-EEG-based emotion recognition. In the proposed method, the input EEG signals undergo a preprocessing stage for noise removal and baseline adjustment. Then, the DWT approach decomposes preprocessed EEG signals into sub-bands (rhythms). Each sub-band extracts the DE features from all channels considered and fed to SVM, kNN, NB, DT, and LR classifiers to find the optimum performing classifier. In this work, we used the VREED dataset obtained by 3D VR videos with positive and negative emotions. We obtained a high average accuracy of 76.22%. To maintain a relatively consistent percentage in each class and each set compared with the original data, hold-out cross-validation was used by splitting each participant’s samples into a training set and a test set. Table 2 compares the obtained results with another reported work [38].

We compared our proposed method’s accuracy with the work by Yu et al. [38]. They provided a baseline for the effectiveness of the classification of negative and positive emotions using the new dataset. They used relative power and mean PLV approaches for feature extraction and SVM classifier to classify the obtained features. The authors reported that combining the relative power characteristics in the theta, alpha, beta, and gamma bands produced the best average accuracy of 73.77%. As seen in Table 2, the obtained average accuracy score of 76.22% is better than the method by Yu et al. [38]. The extracted feature is the main difference between the two works. As mentioned earlier, Yu et al. [38] considered extracting the RP and PLV features to characterize the EEG signals. On the other hand, we used the DE features, a nonlinear method to extract the hidden erratic nature of the EEG signals and discriminate the two emotions.

The advantages of the proposed work are provided below:(1-)The DE feature extraction is a simple method with low computational complexity;(2-)The DE features are nonlinear in nature, hence able to extract hidden complexities in the EEG signals effectively;(3-)The RP and PLV methods used in [38] try to capture regular repeating patterns, while DE provides better results for irregular rhythms such as EEG signals. Moreover, DE is a measure of disorder, and our method obtained high classification performance.

The disadvantages of the proposed study are:
(4-)As mentioned in [46], DE features are unsuitable for CNN-type classifiers.

## 5. Conclusions

This paper studied EEG-based emotion recognition on a virtual reality-based EEG dataset. First, we preprocessed the EEG signals to apply the proposed classification methods. Then, decomposed the preprocessed EEG signals of all channels into four sub-bands and extracted DE features. In the classification step, we used five classifiers, namely SVM, kNN, NB, DT, and LR, to classify into positive or negative valence.

Our results show that the gamma rhythm produced efficient features that yielded high average classification accuracies with most of the classifiers. The DE is a nonlinear feature that is able to extract the hidden irregularity present within the EEG signals which has yielded high classification performance (76.22% ± 2.06).

The test results prove that DE can be used as a suitable measure for evaluating chaotic situations, such as inconsistencies, complexities, and unpredictability, using EEG signals.

The literature [47,48,49,50,51] shows that the high-frequency band of EEG signals, especially the gamma band, has more intense network connections between positive, neutral, and negative moods than the other frequency bands. Hence, this band contributed significantly to high classification performance. The limitation of this work is that classification accuracy is not very high. In the future, we plan to develop deep learning models to increase the model’s performance for the VREED dataset.

In future works, we will investigate the achievement of the deep learning methods on the dataset. Moreover, various feature extraction and selection algorithms can be incorporated to improve the classification accuracy score [52].

## Figures and Tables

**Figure 1 diagnostics-12-02508-f001:**
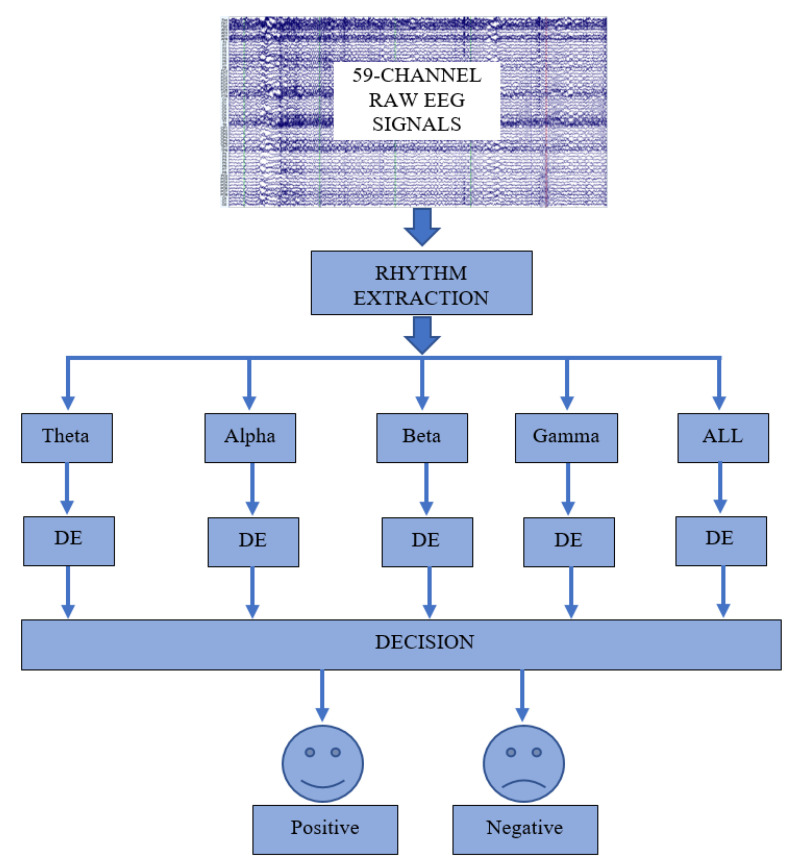
Illustration of the proposed EEG-based emotion recognition system.

**Figure 2 diagnostics-12-02508-f002:**
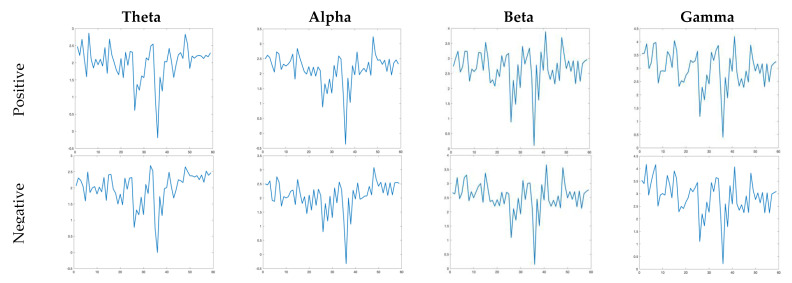
The calculated DE features for all channels of each rhythm for Subject-12 for positive and negative emotions where the *x*-axis (horizontal) shows channel number and the *y*-axis (vertical) shows DE value.

**Figure 3 diagnostics-12-02508-f003:**
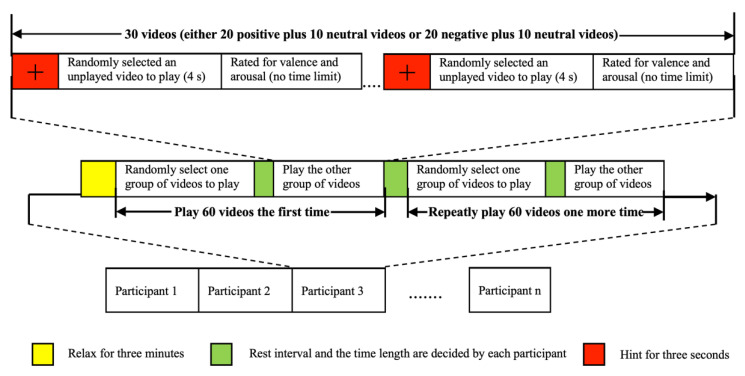
The dataset collection protocol [38].

**Figure 4 diagnostics-12-02508-f004:**
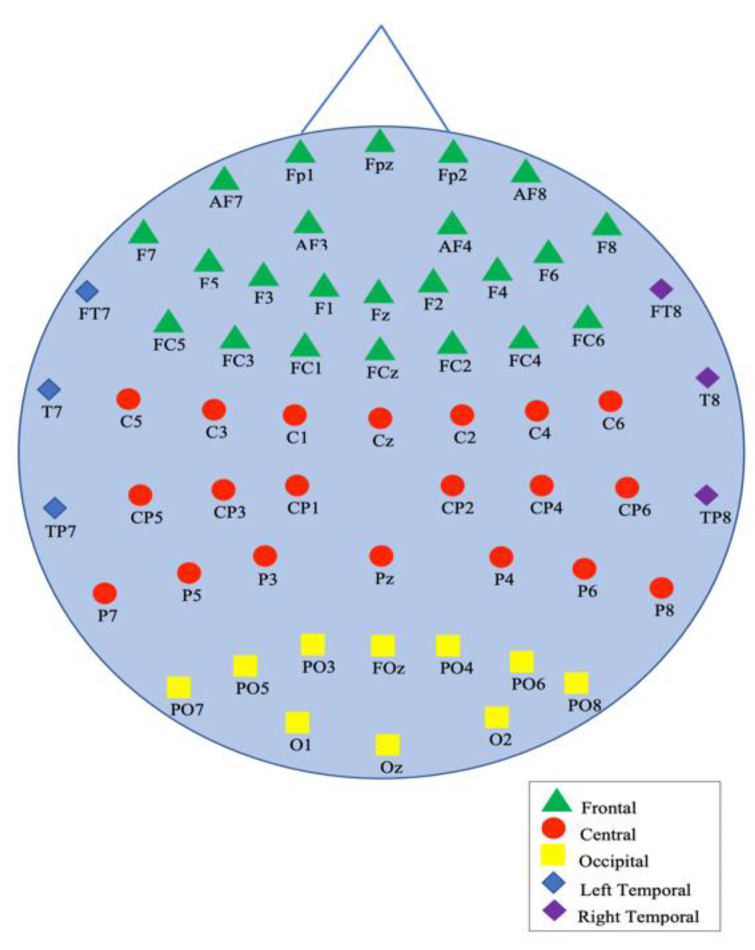
The locations of the electrodes for EEG signal construction [38].

**Figure 5 diagnostics-12-02508-f005:**
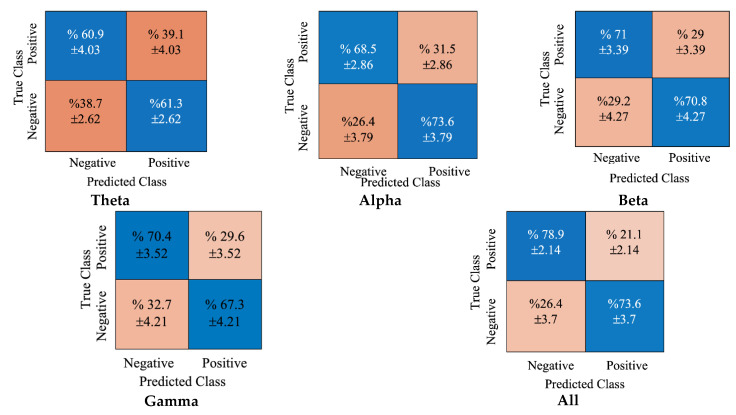
The obtained confusion matrices for DE features and SVM classifier.

**Figure 6 diagnostics-12-02508-f006:**
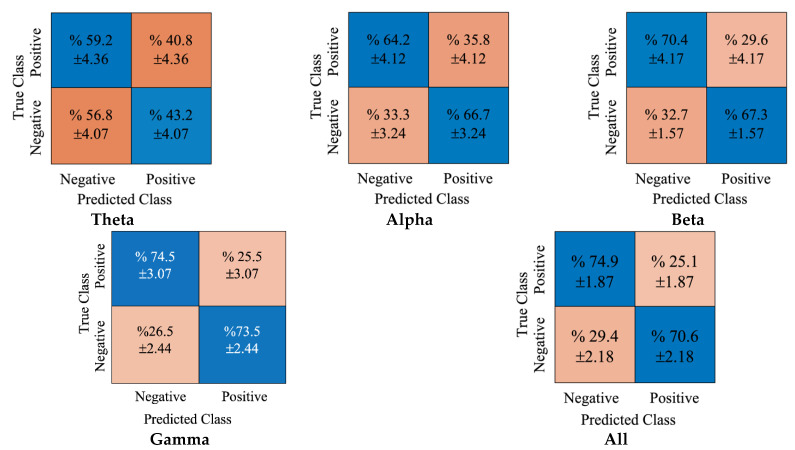
The obtained confusion matrices for DE features and kNN classifier.

**Figure 7 diagnostics-12-02508-f007:**
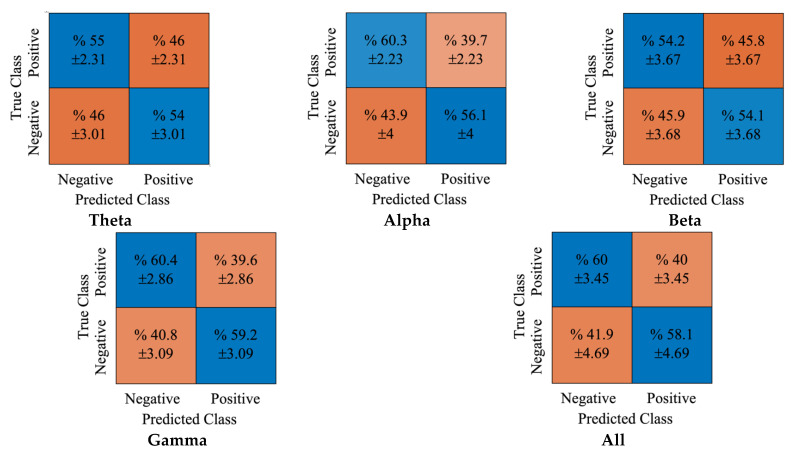
The obtained confusion matrices for DE features and NB classifier.

**Figure 8 diagnostics-12-02508-f008:**
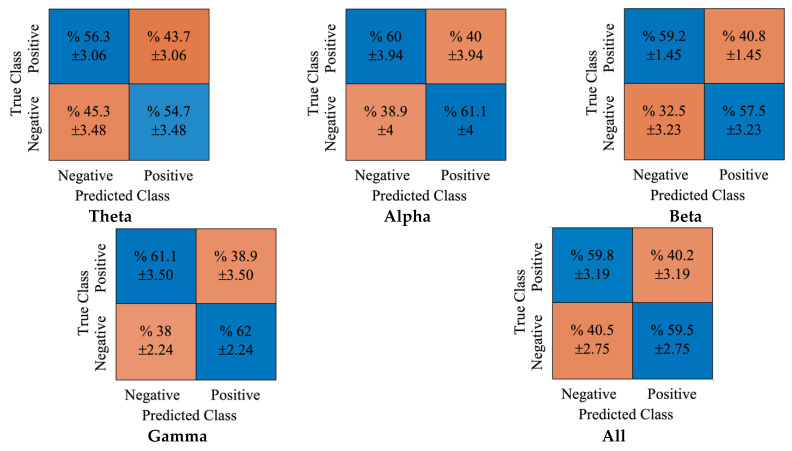
The obtained confusion matrices for DE features and DT classifier.

**Figure 9 diagnostics-12-02508-f009:**
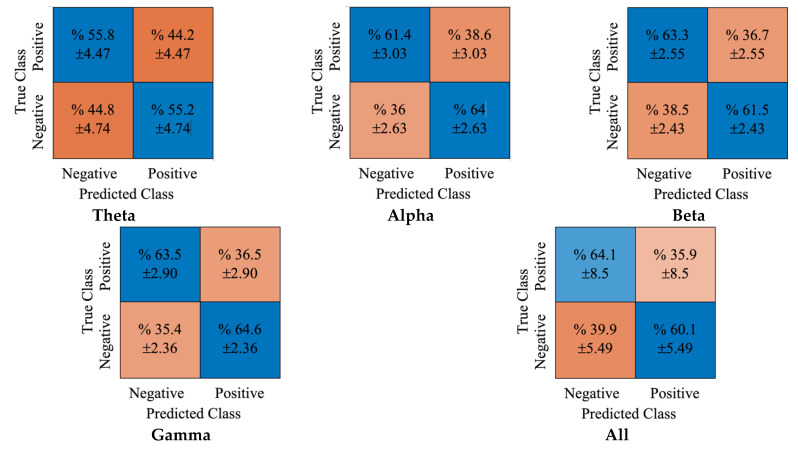
The obtained confusion matrices for DE features and LR classifier.

**Table 1 diagnostics-12-02508-t001:** The achievements of the DE features with SVM, kNN, NB, DT, and LR classifier. The bold cases show the highest average accuracy scores.

	Frequency Bands				
Used Classifier	Theta	Alpha	Beta	Gamma	ALL
SVM	61.0773 ± 2.1893	66.6826 ± 1.4038	70.9104 ± 2.6534	73.9968 ±1.4518	**76.2236 ± 2.0648**
kNN	58.0034 ± 3.6462	65.4013 ± 1.8120	68.8021 ± 2.0587	**72.9261** **± 1.9973**	72.6460 ± 1.1198
NB	54.5211 ± 2.1678	57.9541 ± 2.8655	54.1148 ± 2.7935	**59.7951** **± 1.4314**	59.0151 ± 2.5699
DT	55.5866 ± 2.6421	60.5877 ± 2.1038	58.3750 ± 1.4536	**61.5310** **± 2.8403**	59.6243 ± 2.0847
LR	55.5442 ± 2.8434	62.6286 ± 2.2601	62.4054 ± 1.9679	**64.0111** **± 1.4145**	61.6993 ±3.4722

**Table 2 diagnostics-12-02508-t002:** Comparison of the proposed approach with the existing approach. The bold case indicates the best accuracy score.

Study	Year	Dataset	Extracted Features	Classifier	Validation Method	Highest Accuracy
Yu et al. [38]	2022	VREED	RP	SVM	Hold-out validation (70%–30%) Average of the 10 runs	73.77%
Yu et al. [38]	2022	VREED	MPLV	SVM	Hold-out validation (70%–30%) Average of the 10 runs	67.91%
Proposed	2022	VREED	DE	SVM	Hold-out validation (70%–30%) Average of the 10 runs	76.22%

## Data Availability

Not publicly available.

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
