# Peer review of "Use of Differential Entropy for Automated Emotion Recognition in a Virtual Reality Environment with EEG Signals"

_diagnostics, 2022, doi:10.3390/diagnostics12102508_

Round 1
Reviewer 1 Report
I will list some of my major concerns in the following, which might help the authors to rebuild their current approach.
1. The method does not appear to have a significant improvement in recognition accuracy. The issue of EEG-based emotion recognition achieved more than 80 % classification accuracy at work in 2013, as below.
[1] Differential Entropy Feature for EEG-based Emotion Classification,6th Annual International IEEE EMBS Conference on Neural Engineering San Diego, California, 6 - 8 November, 2013.
More consideration should be given to the accuracy indicators of the existing methods and the differences between the proposed metho, and a detailed explanation should be given.
2. In Table 2, the author only compares their method with the model reported in reference 38(2022). A comparison with one work does not fully illustrate the effectiveness of this approach. The author is advised to add a little more technical comparison.
3. The enumeration numbers on pages 318 to 325 seem problematic. Please correct.
Author Response
Responses to reviewers
We would like to thank the reviewers for his/her precious comments. We tried to fulfill the reviewer's worthy recommendations as well as possible.
The added explanations, changes and other corrections are marked up using the “Track Changes” mode on MS word file in the revised version of the manuscript.
Reviewer 1à
- “The method does not appear to have a significant improvement in recognition accuracy. The issue of EEG-based emotion recognition achieved more than 80 % classification accuracy at work in 2013, as below.
[1] Differential Entropy Feature for EEG-based Emotion Classification,6th Annual International IEEE EMBS Conference on Neural Engineering San Diego, California, 6 - 8 November, 2013.”
Answer: We would like to thank the reviewer for his reference suggestion, where the DE was used for EEG-based emotion classification. However, as we stated in our paper, a new 3D stimuli-based EEG dataset for emotion recognition, namely VREED, was used in our work. So, comparing our paper and the mentioned reference would not be fair.
- “In Table 2, the author only compares their method with the model reported in reference 38(2022). A comparison with one work does not fully illustrate the effectiveness of this approach. The author is advised to add a little more technical comparison.”
Answer: As we mentioned in our paper, the VREED dataset was used in our study and for a convincing comparison, in Table 2, we only put the studies where the VREED dataset was used. To our knowledge, therhas been only one study on the VREED dataset.
- “The enumeration numbers on pages 318 to 325 seem problematic. Please correct.”
Answer: We would like to thank the reviewer for his correction. We handled that problem.

Reviewer 2 Report
The author presents a model to detect sentiments based on EEG. The model looks good however, a few points are critical to improving the paper.
It is recommended to focus only on EEG in the abstract (lines 12-13)
Think of merging the first two sentences in the introduction
"Two kinds of phenomena make up our inner spiritual life. These are thought/cognition and emotion"
Not clear, this refers to which phrase in this sentence. “ A more general and simplified version of this model, a four category structure, is frequently used in the literature"
why dataset [38] was chosen? please provide more details about this dataset in a separate section or as part of the results section
The last two points in the introduction are hard to argue with, maybe you could discuss them in the discussion section. please add a paper outline at the end of the introduction
kindly provide details of the parameters of the machine learning methods in Table 1.
it is critical to discuss why the extracted features of your model (DE) outperform other compared feature models.
please check the format of the text after the line “The advantages of the proposed work are given below:"
You might look for other datasets or models to compare with (if exists). If not it is recommended to explain the findings in more detail showing reasons why the model is better. is it because of the features /model itself?
limitation section is missing
the implication to theory and practice is also missing part. For example, does this model replace generic sentiment models that are available in CNN models?
Author Response
Responses to reviewers
We would like to thank the reviewers for his/her precious comments. We tried to fulfill the reviewer's worthy recommendations as well as possible.
The added explanations, changes and other corrections are marked up using the “Track Changes” mode on MS word file in the revised version of the manuscript
Reviewer 2 à
- “It is recommended to focus only on EEG in the abstract (lines 12-13).”
Answer: The following sentence was added to the abstract for mentioning the importance of the EEG.
‘In particular, EEG signals are bioelectrical signals that are frequently used because of the many advantages they offer in the field of emotion recognition.’
- “Think of merging the first two sentences in the introduction
Answer: According to the reviewers’ comments, those sentences were merged.
"Two kinds of phenomena, namely thought/cognition and emotion, make up our inner spiritual life."
- Not clear, this refers to which phrase in this sentence. “A more general and simplified version of this model, a four category structure, is frequently used in the literature"”
Answer: This conflict has been solved by adding the highlighted word. “multidimensional.”
- “why dataset [38] was chosen? please provide more details about this dataset in a separate section or as part of the results section”
Answer: Thanks to today's technology, EEG datasets created with the help of three-dimensional stimuli are datasets created recently. Such datasets are very important to be able to work in the field of human-computer interaction. Because this area tries to simulate situations that can be closest to our daily life. For this reason, we think that datasets created with the help of three-dimensional stimuli will become much more important in the future. For this reason, we planned to do our work on such a dataset.
The Dataset section is more detailed, as given in Section 2.3
2.3. Dataset
The dataset used in the study was created by watching 4-second long 60 VR videos of three types recorded in 3D by the Shanghai Film Academy with the help of VR glasses. These video types are positive, negative, and neutral. The dataset was initially created with a group of 15 men and 10 women with a mean age of 22.92 and a standard deviation of 1.38 for these ages. The following steps were applied in the emotion elicitation protocol stage:
Figure 3. The dataset collection protocol [38]
As given in Figure 3, 60 videos from three types were randomly selected and divided into two groups, 20 positive - 10 negative and 20 negative - 10 neutral. Researchers who created this dataset did not include negative and positive images in the same video group since VR images have a much higher stimulating effect than normal two-dimensional images. Thus, interference between different emotions is prevented. Each subject was shown two different groups of videos, and the videos in these groups in random order. Participants were given a 3-minute relaxation time before starting the experiment. After a warning sign of 3 seconds, the video was played. In this three-second warning sign, the electrical activity in the brain is marked as a baseline in the dataset. At the end of the videos, a relaxation time was given to the subjects. At the end of each video watched, the subjects were asked to evaluate the induction of the videos on the subjects' emotions. The videos in two groups were shown to each subject twice with the above-mentioned steps. In other words, 120 EEG signal data were collected from each subject.
These recordings were recorded with a 64-channel wireless EEG recorder with a sampling frequency of 1000 Hz, with the impedance of the probes below 5 kΩ. 59 of these 64 channels, corresponding to five different regions of the brain: occipital, parietal, frontal, right temporal, and left temporal, were placed according to the expanded international 10-20 system (10-10 systems) as seen in Figure 4.
Figure 4. The locations of the electrodes for EEG signal construction [38]
After all the recording processes were completed, these data were pre-processed with the help of the EEGLAB Toolbox [39]. Low quality recordings were deleted during this preprocessing. After this deletion, a dataset of 19 people with a mean age of 22.84 and a standard deviation of 1.50 for these ages was formed. Then, the artifacts and power frequency interference of these data were removed and the baseline parts were marked.
- “The last two points in the introduction are hard to argue with, maybe you could discuss them in the discussion section. please add a paper outline at the end of the introduction”
Answer :We could not understand what the reviewer mentioned in his/her suggestion.
à Paper outline:
The rest of the paper is organized as following. In the Section 2, the proposed scheme and its application steps are given. Besides, the theory of the considered methods and the description of the dataset are given in Section 2. In Section 3, the experimental results are interpreted in details. In Section 4, the advantages and disadvantages of the proposed method are emphasized. Finally, the paper is concluded in Section 5.
- “kindly provide details of the parameters of the machine learning methods in Table 1.”
Answer: Actually, a hyperparameter optimization was used for each classifier to tune of the related parameters.
- it is critical to discuss why the extracted features of your model (DE) outperform other compared feature models.
Answer: The mentioned clarification was justified in the conclusion section of the work, as given below;
“Our results show that the gamma rhythm produced the efficient features that have yielded high average classification accuracies with most of the classifiers. On the other hand, the DE is a nonlinear feature that is able to extract the hidden irregularity present within the EEG signals which has yielded high classification performance (76.22% ± 2.06).
”
- “please check the format of the text after the line “The advantages of the proposed work are given below:"”
Answer: We handled that problem.
- You might look for other datasets or models to compare with (if exists). If not it is recommended to explain the findings in more detail showing reasons why the model is better. Is it because of the features /model itself?
Answer: In our future works, we will use more related datasets to improve performance.
- “limitation section is missing”
Answer: We gave the limitations in the discussion and conclusions sections.
- the implication to theory and practice is also missing part. For example, does this model replace generic sentiment models that are available in CNN models?
Answer: We would like to thank the reviewer. Actually, we are working on it now for our future works.

Round 2
Reviewer 1 Report
The authors have already answered my questions well and made the corresponding revisions and updates in their manuscript. I currently recommend this paper for publication in Diagnostics.